# Relationship between a Self-Reported History of Depression and Persistent Elevation in C-Reactive Protein after Myocardial Infarction

**DOI:** 10.3390/jcm11092322

**Published:** 2022-04-21

**Authors:** Hannes Bielas, Rebecca E. Meister-Langraf, Jean-Paul Schmid, Jürgen Barth, Hansjörg Znoj, Ulrich Schnyder, Mary Princip, Roland von Känel

**Affiliations:** 1Department of Child and Adolescent Psychiatry, Technische Universität Dresden, 01307 Dresden, Germany; 2Department of Consultation-Liaison Psychiatry and Psychosomatic Medicine, University Hospital Zürich, 8091 Zurich, Switzerland; rebem1@yahoo.de (R.E.M.-L.); mary.princip@usz.ch (M.P.); roland.vonkaenel@usz.ch (R.v.K.); 3Faculty of Medicine, University of Zürich, 8901 Zurich, Switzerland; ulrich.schnyder@access.uzh.ch; 4Clienia Schlössli AG, 8618 Oetwil am See, Switzerland; 5Department of Cardiology, Clinic Barmelweid, 5017 Barmelweid, Switzerland; jean-paul.schmid@barmelweid.ch; 6Institute for Complementary and Integrative Medicine, University Hospital Zürich, University of Zürich, 8091 Zurich, Switzerland; juergen.barth@usz.ch; 7Department of Clinical Psychology and Psychotherapy, University of Bern, 3012 Bern, Switzerland; hansjoerg.znoj@psy.unibe.ch

**Keywords:** cardiovascular disease, inflammation, psychobiology, risk factor, traumatic stress, depression

## Abstract

Background: Elevated levels of C-reactive protein (CRP) are associated with both an increased risk of cardiovascular disease (CVD) and depression. We aimed to test the hypothesis that a self-report history of depression is associated with a smaller decrease in CRP levels from hospital admission to 3-month follow-up in patients with acute myocardial infarction (MI). Methods: We assessed 183 patients (median age 59 years; 84% men) with verified MI for a self-report history of lifetime depression and plasma CRP levels within 48 h of an acute coronary intervention and again for CRP levels at three months. CRP values were categorized according to their potential to predict CVD risk at hospital admission (acute inflammatory response: 0 to <5 mg/L, 5 to <10 mg/L, 10 to <20 mg/L, and ≥20 mg/L) and at 3 months (low-grade inflammation: 0 to <1 mg/L, 1 to <3 mg/L, and ≥3 mg/L). Additionally, in a subsample of 84 patients showing admission CRP levels below 20 mg/L, changes in continuous CRP values over time were also analyzed. Results: After adjustment for a range of potentially important covariates, depression history showed a significant association with a smaller decrease in both CRP risk categories (r = 0.261, *p* < 0.001) and log CRP levels (r = 0.340, *p* = 0.005) over time. Conclusions: Self-reported history of depression may be associated with persistently elevated systemic inflammation three months after MI. This finding warrants studies to test whether lowering of inflammation in patients with an acute MI and a history of depression may improve prognosis.

## 1. Introduction

Low-grade systemic chronic inflammation, measured by circulating levels of C-reactive protein (CRP), has been identified as a cardiovascular risk factor [1]. As such, CRP has been associated with incident cardiovascular disease (CVD), including coronary heart disease (CHD), in multiple studies [2]. Based on data from prospective cohort studies, categories of low, moderate, and high cardiovascular risk can be predicted from CRP levels of <1, 1 to <3, and ≥3 mg/L, respectively [3]. Moreover, CRP risk-categories, such as <5, 5 to <10, 10 to <20, and ≥20 mg/L differentially extend the prognostic information above and beyond the Framingham risk score for CHD [4].

We previously showed a high CRP risk-category between hospital admission (CRP levels reflecting acute inflammatory response: <5, 5 to <10, 10 to <20, and ≥20 mg/L), and three-month follow-up (CRP levels reflecting systemic low-grade inflammation: <1, 1 to <3, and ≥3 mg/L) to predict the risk of post-traumatic stress disorder symptoms (PTSS) independent of sociodemographic, psychosocial, and medical factors [5]. Moreover, in this previous study, the use of antidepressant medication at time of admission was more than four times higher in the group of patients with clinically relevant MI-induced PTSS, indicating a possible role of a previously diagnosed episode of depression [5]. Whereas the association of proinflammatory changes with PTSS has been shown to be independent of depression [6], depression has also emerged as an independent risk factor for incident and recurrent cardiovascular events and mortality in addition to inflammation and traditional cardiovascular risk factors [7,8,9,10,11]. There is therefore a complex relation between depression and CHD regardless of comorbidity with PTSS, but how this may be relevant to persistently increased CRP risk-categories after MI remains to be further investigated.

In an attempt to broaden this knowledge, the aim of this study was to test the hypothesis of a direct relationship between a history of lifetime depression and persistent elevation in CRP across risk categories between hospital admission due to MI (when CRP levels reflect an acute phase response) and three months later (when CRP levels reflect low-grade inflammation). We specifically hypothesized that the relationship between a history of depression and persistent elevation of CRP (with levels categorized by CVD risk) between hospital admission and three months later would be independent of sociodemographic factors, health behaviors, cardiac disease-related variables, use of antidepressants and cortisol at admission. The latter was taken into account, as it has been shown that a dysfunctional hypothalamic-pituitary adrenal axis response in patients with an acute coronary syndrome involves a failure to confine inflammatory activity [12].

## 2. Materials and Methods

### 2.1. Patients and Study Design

Between January 2013 and September 2015, the Myocardial Infarction-Stress Prevention Intervention (MI-SPRINT) study recruited a sample of 190 eligible patients from the Bern University Hospital (“Inselspital”) referred for acute coronary care intervention due to verified acute ST-elevation MI (STEMI) or non-STEMI. The trial protocol was approved by the ethics committee of the State of Bern, Switzerland (ClinicalTrials.gov; NCT01781247; http://clinicaltrials.gov/show/NCT01781247, accessed on 10 November 2021), and conforms to the ethical guidelines of the 1975 Declaration of Helsinki. MI-SPRINT was a randomized controlled trial with the aim to prevent the development of interviewer-rated PTSS after 3 months through trauma-focused counseling compared with stress-focused counseling (active control) both delivered as a single session to MI patients at hospital admission [13]. The elements of both counseling sessions have been detailed elsewhere, along with the main finding of the MI-SPRINT trial that trauma-focused counseling and stress counseling resulted in similar interviewer-rated PTSS severity 3 months post-MI [14].

Patients provided signed informed consent to the study, which was approved by the ethics committee of the State of Bern, Switzerland (KEK-Nr. 170/12). Included were participants 18 years or older with a substantial level of acute distress perceived during MI (score of at least 5 for chest pain and for fear of dying and/or helplessness on a numeric rating scale from 0–10) [15]. After having reached stable hemodynamic conditions, all patients underwent a structured clinical interview within 48 h during which a medical history, including medication use, and information on health behaviors was collected. Patients also completed self-rated psychometric questionnaires. Fasting venous blood samples were drawn the next morning for the assessment of CRP (in less than 10% of cases blood was collected at another time of the day and non-fasting due to logistical reasons). Exclusion criteria were emergency coronary artery bypass grafting, any severe disease likely to cause death within one year, cognitive impairment, a current severe depressive episode (per the cardiologists’ clinical judgement), suicidal ideation in the last two weeks, insufficient German language proficiency, or participation in another randomized controlled trial (see Figure 1). At 3-month follow-up, plasma CRP levels were determined again.

### 2.2. Measures

C-reactive protein: The dependent variable CRP was measured in lithium-heparin plasma with an immunoturbidimetric assay (C-Reactive Protein Gen.3, measuring range 0.3–350 mg/L) using the COBAS 8000 c702 module from Roche Diagnostics. The assay was performed according to the manufacturer’s instructions at the Central Laboratory for Clinical chemistry-CoreLab, Bern University Hospital, Bern, Switzerland. Circulating levels of CRP at hospital admission (acute inflammatory response) were categorized according to the prognostic risk of CRP levels for incident CVD as follows: 0 to <5 = 1, 5 to <10 = 2, 10 to <20 = 3, and ≥20 mg/L = 4 [4]. The cut points for the risk categorization at follow-up (low-grade inflammation) in terms of low risk = 1 (<1.0 mg/L), average risk = 2 (1.0 to 3.0 mg/L), and high risk = 3 (>3.0 mg/L) quite correspond with tertiles of CRP in the adult population [3]. The change of CRP categories (ΔCRP cat) was calculated by subtracting the classified risk category at admission from the classified risk category at follow-up.

Independent variables: Depression history in terms of an episode of lifetime depression was by patients’ self-report. Specifically, patients were asked the following question with a “yes” or “no” response format: “Have you ever had a depression in your life?”. Serum cortisol was analyzed by an electro-chemiluminescence immunoassay on a Cobas analyzer (Roche Diagnostics, Switzerland) in the CoreLab, Bern University Hospital, Switzerland [16]. Perceived social support was assessed with the Enhancing Recovery in CHD Patients Social Support Inventory (ESSI), comprising dimensions of emotional, structural and instrumental support, with six items rated on a Likert scale from 0 (“none of the time”) to 4 (“all the time”) (Mitchell et al., 2003 [17]). Socioeconomic status was defined with reference to a high, medium, or low level of education [18]. Participants disclosed their weight and height for the calculation of the body mass index (BMI). We also assessed smoking habits (current, former or never smokers), level of physical activity (“that makes you sweat”) in terms of the number of times in an average week, and consumption of alcoholic beverages. According to the well-known J-shaped risk between alcohol intake and CVD risk [19], we categorized participants on a scale from 0–2 as moderate drinkers, non-drinkers, and heavy drinkers (>21 drinks/week for men, >14 drinks/week for women). Cardiac disease-related measures were type of MI, either ST-elevation MI (STEMI) or non-STEMI [20], and the Global Registry of Acute Coronary Events (GRACE) risk score, which combines eight variables to estimate the risk of post-discharge death and recurrent MI after acute coronary syndrome [21]. Left ventricular ejection fraction (LVEF) was obtained from angiographic records, and white blood cell (WBC) count and peak troponin T levels were abstracted from hospital records.

### 2.3. Statistical Analyses

Data were analyzed using SPSS 26.0 for Windows (SPSS Inc., Chicago, IL, USA) with level of significance at *p* < 0.05 (two-sided). As shown in Figure 1, of the initially enrolled 190 participants, seven had died at the time of the follow-up, so we analyzed follow-up data from 183 patients. CRP was not measured in 17 (9.3%) cases at admission and in 63 (34.4%) cases at follow-up. Information on depression history was lacking in 3 (1.6%) cases. The GRACE risk score could not be computed for 18 (9.8%) patients; 43 (23.5%) cases missed social support indications. Four or less values were missing for the other measures. We replaced all missing values with the expectation maximization algorithm to make use of all the available information from the total sample of 183 study participants. Little’s missing completely at random (MCAR) tests revealed no significant patterns before performing imputations.

We used Pearson’s chi-squared test and Fisher’s exact test where appropriate, to compare differences of patient characteristics between whole sample and subgroup. Pearson correlation analyses was used to estimate the relationship between two variables. Multivariable linear regression analyses were used to examine the independent relationship between depression history and the change of CRP risk categories (Δcat-CRP, range −3 to 2) from hospital admission to 3-month follow-up. Covariate adjustment was made for sociodemographic factors (age, gender, education), health behaviors (BMI, smoking, alcohol consumption, physical activity), cardiac-related variables (type of MI, LVEF, GRACE risk score, peak troponin, WBC count), social support, antidepressant medication, cortisol and type of psychological counseling intervention (all variables entered in one block). We selected the covariates a priori based on the literature, as they might potentially confound associations with CRP levels and due to a potential intervention effect of the type of counseling (trauma-focused vs. stress counseling). We allowed a maximum of 17 independent variables to prevent the model from being overfitted. As the core lab indicated no discrete values for CRP levels ≥20 mg/L, we conducted an additional analysis for participants with CRP-levels less than 20 mg/L with log (base 10) transformed CRP values to achieve a linear distribution of CRP values.

## 3. Results

### 3.1. Patient Characteristics

Table 1 shows the characteristics of the 183 patients with acute MI who participated in the 3-month follow-up investigation. All were of Caucasian ethnicity. The first column depicts the data of the 84 participants with CRP-levels below 20 mg/L at admission and the second column displays the data (imputed when necessary) of the full sample. The second column shows mean values or counts and percentages of patient characteristics for each Δcat-CRP in the imputed data set considering all 183 participants. The subgroup with 84 participants did not significantly differ in characteristics from the total sample of 183 participants. The majority of participants were well-educated men. A history of depression was reported in almost 30% and antidepressant medication was used in less than 10% of cases. According to the GRACE score, the median (inter-quartile range) risk to die within the next 6 months was 5% (3–9). Regarding health behaviors, patients were on average overweight, and a substantial portion were current smokers and physically inactive. Regarding CRP risk categories, 42 (23.0%) participants showed no decrease and 8 (4.4%) showed an increase in the CRP risk category from admission to 3-month follow-up.

### 3.2. Unadjusted Associations of Depression History and Covariates with the Course of CRP

As shown in Table 1, there were several significant zero-order correlations with Δcat-CRP. Δcat-CRP was associated with a history of depression (r = 0.231, *p* = 0.001), STEMI (r = 0.196, *p* = 0.004), female gender (r = 0.201, *p* = 0.003), BMI (r = −0.144, *p* = 0.026), physical activity (r = 0.153, *p* = 0.020), GRACE risk score (r = 0.193, *p* = 0.004), LVEF (r = 0.320, *p* ≤ 0.001), ESSI (r = −0.126, *p* = 0.045), WBC count (r = −0.246, *p* < 0.001), cortisol (r = −0.231, *p* = 0.001), and troponin (r = −0.205, *p* = 0.003). For the subgroup of 84 patients, significant Pearson’s correlations were found between Δlog CRP values and depression history (r = 0.299, *p* = 0.003), troponin (r = −0.230, *p* = 0.018), and antidepressant use (r = 0.385, *p* < 0.001).

### 3.3. Independent Association between Depression History and the Course of CRP

The results of the two multivariable linear regression analyses with all variables entered in one block are shown in Table 2. The models explained 22% of the variance in the shift of Δcat-CRP and 27% of the variance in the change of log CRP levels in the subgroup of 84 participants without excessive CRP responses during acute MI. History of depression was an independent predictor of Δcat-CRP (β = 0.218, t(165) = 3.478, *p* < 0.001), such that patients with lifetime depression showed little decrease in Δcat-CRP compared to patients without lifetime depression. A similar effect was observed for the change in log CRP levels (β = 0.323, t(66) = 2.940, *p* = 0.005) independent from type of MI, as illustrated in Figure 2. Regarding covariates in the models, as can also be seen in Table 2, little decrease in Δcat-CRP was significantly predicted by lower BMI and higher LVEF. In addition, lower troponin levels and use of antidepressant medication were independent predictors of a smaller decrease in log CRP levels over time.

## 4. Discussion

We examined the predictive value of a history of depression in 183 patients with acute MI at 3-month follow-up for the persistence of initially increased CRP, which is known to be a risk factor for CVD [1]. We evaluated the change over time in both clinically established risk categories of CRP and log CRP levels. The latter was done in a subgroup of 84 patients with CRP levels below 20 mg/L at hospital admission, allowing us to examine associations in patients with only a moderate acute phase response. Adjusting for a range of covariates, we identified lifetime depression before MI onset as an independent predictor of a smaller decrease over time in Δcat-CRP and Δlog CRP with similar effect sizes.

Paradoxically, lower BMI and higher LVEF also showed a significantly smaller decrease in Δcat-CRP over time, although this was not observed in the analysis in the subsample with Δlog CRP. These contrasting results may be interpreted as a potentially greater regression to the mean effect in patients with CRP levels exceeding 20 mg/L [22]. This explanation could also apply to the unexpected zero-order correlations between a low reduction in Δcat-CRP and lower BMI, WBC count, cortisol levels and peak troponin on the one hand and increased physical activity on the other. Except troponin, these variables showed no significant associations with changes in log CRP. Moreover, these observations may be attributed to greater recovery from an acute inflammatory response in initially more affected participants [22]. Our study was not designed to examine direct pathophysiological mechanisms that might link a history of depression with CRP levels in patients with acute MI. However, dysfunction in the autonomic nervous system (e.g., vagal withdrawal) [23] and the hypothalamic-pituitary adrenal axis (e.g., low cortisol response) [12] are candidate mechanisms, both of which may result in failure to confine inflammatory activity.

None of the examined risk factors of CVD, but a positive history of depression was consistently identified as a significant predictor of persistently increased CRP levels in both fully adjusted regression models. This finding adds to the literature demonstrating that CHD patients with depression had greater impairment in myocardial perfusion when acutely being stressed [24]. The assumption of a smaller decrease in Δcat-CRP potentially contributing to such an association is in agreement with a recent study that showed elevated CRP in the acute phase of MI to be a predictor of major adverse cardiovascular events at 1-year follow-up [25]. The authors reported an increase in CVD risk across risk categories with higher levels of CRP at admission ranging from below 2 mg/L to above 10 mg/L. Consequently, our data suggest that little decrease in CRP levels after MI could put patients with lifetime depression at risk for recurrent MI. In other words, persistently higher CRP risk-categories and log CRP levels may explain the previously reported link of premorbid depression and poorer prognosis of patients with CHD [26]. This may prompt researchers to have a closer look into future studies on a history of lifetime depression as a potentially independent prognostic factor in patients with acute coronary syndromes. Although we included patients with classical MI due to coronary atherosclerotic disease, it should be mentioned that patients with depression are prone to MI with non-obstructive coronary arteries which is related to microvascular dysfunction predisposing to ischemic heart disease [27,28].

Addressing a history of depression as a risk factor of persistently elevated CRP levels may also be considered to prevent the development of MI-induced PTSS [5]. That is, as the present sample was recruited based on high levels of MI-triggered peritraumatic distress, our findings may have particular implications for the association between CRP and PTSS, particularly for the treatment of individuals with high threat reactivity [29].

Antidepressants are largely ineffective in improving disease outcome in different populations of patients with CHD [30]. Our finding of a positive relationship between the use of antidepressant and less of a decrease in CRP levels in patients with CRP levels below 20 mg/L at admission concurs with the notion that antidepressants have limited benefit in “inflamed patients”. For instance, an earlier study in patients with treatment-resistant depression showed that the tumor necrosis factor antagonist infliximab reduced depressive symptoms to a greater extent than placebo when baseline CRP levels were above 5 mg/L compared to when CRP levels were below this threshold [31]. Meanwhile, there is accumulated evidence from several meta-analysis that in patients with depression, anti-inflammatory treatments, including anti-inflammatory agents and anti-cytokine therapy, significantly reduced depressive symptoms compared to placebo [32,33,34]. A parallel line of research has shown that anti-inflammatory therapy with the interleukin-1β inhibitor canakinumab resulted in a reduced risk of major adverse cardiovascular events in patients with prior MI and residual inflammatory risk (CRP levels ≥ 2 mg/L) [35]. Although not currently approved for cardiovascular indications, other clinically available interleukin-1 inhibitors as well as oral NLRP3 inflammasome inhibitors under development, also promise to improve outcomes in patients with acute MI [36]. It is an intriguing assumption that based on this literature patients with a history of depression and elevated CRP post-MI might particularly benefit from, for instance, anti-interleukin-1 treatment in terms of better cardiovascular outcome. Therefore, the issue of a potentially prognostic benefit of anti-inflammatory treatment in the aftermath of a coronary event in patients with a history of depression may warrant further investigation.

Our study has several limitations. First, the data are from a secondary analysis of a randomized controlled behavioral intervention trail [13]. The hypothesis tested here was not pre-specified in the original study protocol, and thus it is exploratory. As a consequence, inflammatory diseases were not an exclusion criterion. Second, the measurement of CRP changes over time cannot directly be translated to CRP categories [37], but it provides sufficient evidence to develop a potential biomarker for clinical evaluation in the follow-up of distressed MI patients. Third, the current results should not be generalized to samples with a higher share of women, less educated patients, and those with lower levels of peritraumatic distress. Fourth, covariates were selected based a priori assumptions on associations with CRP levels and were limited in number to avoid unstable coefficients due to model overfitting, so residual confounding by variables which were not controlled for remains a possibility. For instance, we were unable to additionally control in regression models for diabetes, prior heart failure, prior MI, and elevated triglycerides and blood pressure [38]. Similarly, only LVEF at the time of acute MI was available to be included as a covariate, although other cardiac-disease related measures have also been shown to be associated with persistent elevation in CRP concentration after MI, including angiographic indicators of reperfusion success, therapy, post-MI left ventricular dysfunction and remodeling, and heart failure [39,40]. Fifth, depression history was assessed by patients’ self-reported recall of remote depressive episodes, which can be biased by their current psychological state and medical condition. A structured clinical interview to assess the severity and duration of individual depressive symptoms for a definition of a major depressive episode according to criteria of the Diagnostic and Statistical Manual of Mental Disorders would have yielded greater confidence in the presence or absence of an episode of lifetime depression.

## 5. Conclusions

To sum up, our data suggest that a history of depression may play a role in the persistence of increased CRP levels in patients after MI, pointing toward a potentially increased risk of recurrent CVD events and mortality. Larger studies allowing for extensive adjustment of correlates of CRP post-MI may further substantiate an independent relationship between lifetime depression and persistently elevated CRP levels post-MI. Nonetheless, knowledge on the shift in CRP-related CVD-risk categories related to lifetime depression can be of clinical use to identify patients at need for tailored anti-inflammatory interventions in the aftermath of MI. To inform such interventions, future studies are needed to evaluate the clinical relevance of changes in CRP for the cardiac prognosis in patients with lifetime depression.

## Figures and Tables

**Figure 1 jcm-11-02322-f001:**
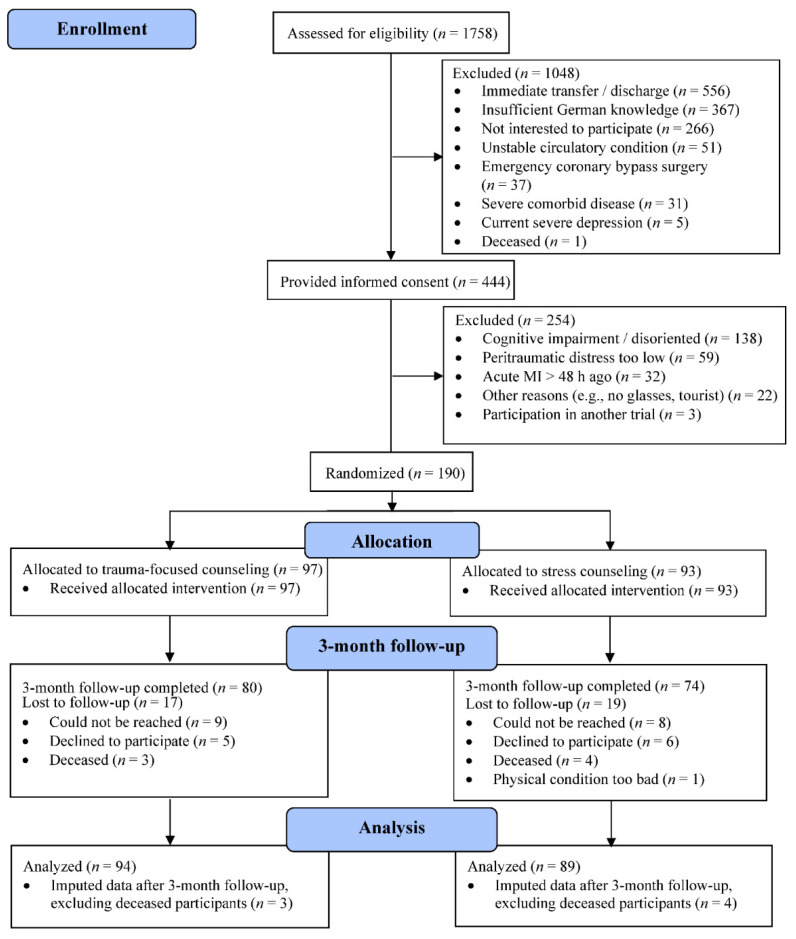
Consort flow diagram for 3-month follow-up participants (*n* = 183) of the MI-SPRINT trial.

**Figure 2 jcm-11-02322-f002:**
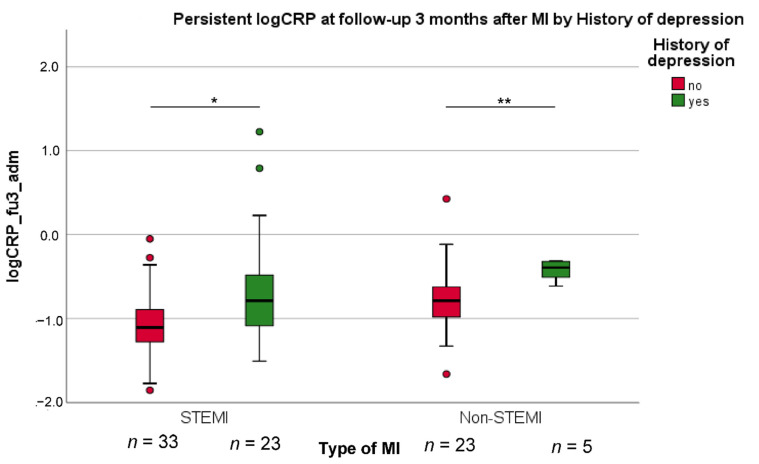
The decrease of high log CRP values at 3-month follow-up compared to admission after myocardial infarction (MI) (fu3_adm), in both the group with ST-elevation MI (STEMI) and with non-STEMI. *t*-test for significant group differences: * *p* < 0.05; ** *p* < 0.01.

**Table 1 jcm-11-02322-t001:** Characteristics of patients with 3-month follow-up after acute myocardial infarction according to change in cardiovascular risk categories.

	Δ log CRP	Δ Cat CRP (CRP Risk-Category at Follow-Up Minus CRP Risk-Category at Admission)						
		All	−3	−2	−1	0	1	2
N	84	183	21	54	66	31	9	2
Age, yrs, M (SD)	58.8 (10.7)	59.3 (10.9)	59.6 (9.7)	60.8 (11.1)	58.4 (12.1)	58.0 (9.2)	60.1 (9.2)	63.5 (16.3)
Sex **								
Female, *n* (%)	19 (22.6)	**29 (15.8)**	0 (0)	7 (1.8)	11 (20)	8 (35.3)	2 (30)	1 (50)
Male, *n* (%)	65 (77.4)	**154 (84.2)**	21 (100)	47 (98.2)	55 (80)	23 (64.7)	7 (70)	1 (50)
Intervention								
Trauma-focused, *n* (%)	45 (53.6)	94 (51.4)	9 (43)	28 (52)	32 (48)	19 (61)	4 (44)	2 (100)
Stress-focused, *n* (%)	39 (46.4)	89 (48.6)	12 (57)	26 (48)	34 (52)	12 (39)	5 (56)	0 (0)
Social support, M (SD) *	19.3 (4.0)	**19.7 (4.1)**	20.9 (3.3)	20.0 (4.6)	19.4 (4.1)	19.1 (3.8)	19.9 (3.0)	16.5 (3.5)
History of depression *n* (%) **	28 (33.3)	**52 (28.4)**	4 (19)	8 (15)	23 (35)	11 (35)	4 (44)	2 (100)
Education level								
High, *n* (%)	21 (25)	34 (18.6)	3 (14)	10 (19)	9 (14)	2 (6)	2 (22)	2 (100)
Medium, *n* (%)	58 (69)	131 (71.6)	17 (81)	37 (69)	49 (74)	21 (68)	7 (78)	0 (0)
Low, *n* (%)	5 (6.0)	18 (9.8)	1 (5)	7 (13)	8 (12)	8 (26)	0 (0)	0 (0)
Body mass index, kg/m^2^, M (SD) *	26.9 (3.9)	**27.8 (4.5)**	27.1 (3.7)	29.1 (4.6)	28.0 (4.7)	26.1 (4.7)	26.5 (3.1)	26.0 (3.1)
Smoking status								
Current smoker, *n* (%)	34 (40.5)	80 (43.7)	8 (38)	22 (36.8)	37 (41)	11 (35)	2 (22)	0 (0)
Former smoker, *n* (%)	24 (28.6)	49 (26.8)	6 (29)	20 (33.3)	9 (17)	8 (26)	5 (56)	1 (50.0)
Never smoker, *n* (%)	26 (31)	54 (29.5)	7 (33)	12 (29.8)	20 (37)	12 (39)	2 (22)	1 (50.0)
Alcohol consumption								
Moderate drinkers, *n* (%)	60 (71.4)	131 (71.6)	15 (71)	38 (70)	47 (71)	23 (74)	7 (78)	1 (50.0)
Abstainers, *n* (%)	15 (17.9)	33 (18.0)	3 (14)	8 (15)	16 (24)	4 (13)	1 (11)	1 (50.0)
Heavy drinkers, *n* (%)	9 (10.7)	19 (10.4)	3 (14)	8 (15)	3 (5)	4 (13)	1 (11)	0 (0)
Physical activity (times per week) *								
3–7, *n* (%)	26 (36.9)	**48 (26.2)**	6 (29)	8 (15.8)	20 (30)	9 (29)	4 (44)	1 (50)
1–2, *n* (%)	27 (32.1)	**51 (27.9)**	4 (19)	17 (31.6)	15 (23)	13 (42)	2 (22)	0 (0)
<1, *n* (%)	31 (36.9)	**84 (45.9)**	11 (52)	29 (52.6)	31 (47)	9 (29)	3 (33)	1 (50)
ST-elevation MI **								
Yes, *n* (%)	56 (66.6)	**132 (72.1)**	18 (86)	43 (80)	47 (71)	18 (58)	4 (44)	2 (100)
No, *n* (%)	28 (33.3)	**51 (27.9)**	3 (14)	11 (20)	19 (29)	13 (42)	5 (56)	0
White blood cell count, ×10^9^/L, M (SD) ***	8.4 (1.7)	**9.1 (2.4)**	10.3 (3.9)	9.4 (1.9)	9.3 (2.2)	7.9 (1.7)	8.3 (2.3)	9.9 (2.1)
Troponin T peak level, μg/L, M (SD) **	1.2 (2.6)	**1.8 (3.8)**	2.5 (4.6)	2.9 (5.3)	1.5 (2.7)	0.4 (0.8)	0.5 (0.9)	2.5 (1.9)
GRACE risk score, M (SD) **	100 (23.3)	**105 (25.7)**	113.7 (25.5)	110 (25.0)	103.5 (27.2)	95.9 (21.9)	97.4 (22.0)	117 (27.6)
LVEF, %, M (SD) ***	51.0 (10.6)	**47.6 (11.5)**	44.1 (16.8)	42.6 (9.4)	48.7 (9.6)	55.5 (9.7)	54.4 (10.4)	37.5 (3.5)
Cortisol at admission, nmol/L, M (SD) **	473(165)	**501 (184)**	569 (248)	541 (161)	493 (184)	415 (164)	450 (107)	535 (23)
Antidepressant use, *n* (%)	4 (4.8)	13 (7.1)	1 (5)	2 (4)	6 (9)	2 (6)	2 (22)	0 (0)

Data are given as mean (M) with standard deviation (SD) or numbers (*n*) with percentage value (%). Δ cat CRP, change in risk category of C-reactive protein; Δ log CRP, change in log CRP; GRACE, Global Registry of Acute Coronary Events; LVEF, left ventricular ejection fraction; MI, myocardial infarction. Significant associations of patient characteristics with changes in Δ log CRP and Δ cat CRP from admission to follow-up are displayed in **bold**: * *p* < 0.05; ** *p* < 0.01; *** *p* ≤ 0.001.

**Table 2 jcm-11-02322-t002:** Multivariable relationship between depression history and CRP change.

Entered Variables (One Block)	Δ Cat CRP		Δ log CRP	
	Partial Corr.	P	Partial Corr.	P
Trauma focused intervention	−0.029	0.715	0.035	0.778
Social support	−0.014	0.861	0.004	0.973
Female gender	0.076	0.326	−0.101	0.411
Age	−0.049	0.528	0.140	0.255
Body mass index **	**−0.202**	**0.009**	−0.034	0.785
Education	0.073	0.351	0.219	0.073
Smoking	−0.036	0.640	−0.158	0.199
Alcohol consumption	−0.039	0.613	−0.061	0.623
Physical activity	0.138	0.075	−0.123	0.318
GRACE risk score	−0.027	0.730	−0.088	0.474
LVEF *	**0.172**	**0.027**	−0.022	0.860
Non-ST-elevation MI	0.132	0.089	0.219	0.072
White blood cell count	−0.069	0.376	0.239	0.050
Cortisol at admission	−0.118	0.129	0.170	0.166
Troponin	−0.106	0.174	**−0.257**	**0.034**
Antidepressants	0.057	0.464	**0.391**	**<0.001**
Depression history ***	**0.261**	**<0.001**	**0.340**	**0.005**
Model statistics	R^2^ = 0.22 F_17,165_ = 4.02 *p* < 0.001		R^2^ = 0.27 F_17,66_ = 2.81 *p* = 0.001	

Data are given as partial correlation (corr) coefficient. Δ cat CRP, change in risk category of C-reactive protein; Δ log CRP, change in log CRP; GRACE, Global Registry of Acute Coronary Events; MI, myocardial infarction. Significant associations with changes in Δ cat CRP and Δ log CRP from admission to follow-up are displayed in **bold**: * *p* < 0.05; ** *p* < 0.01; *** *p* ≤ 0.001.

## Data Availability

Data available on request due to privacy restrictions.

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
