# Peer review of "Relationship between a Self-Reported History of Depression and Persistent Elevation in C-Reactive Protein after Myocardial Infarction"

_jcm, 2022, doi:10.3390/jcm11092322_

Round 1

Reviewer 1 Report

The article is interesting and it focuses on the association between persistent elevated CRP in pateints with myocardial infarction and an history of depression. Some aspects should be improved: 

1) There is a lack of a real control group. How can you affirm that patients with depression have a slower decline of CRP compared to patients without depression?

2) add an algoritm with study design

3) depressive disorders are a major risk factor for microvascular dysfunction and MINOCA. Please discuss the role of microcirculation in ischemic heart disease predisposition (see J cardiovasc Dev Dis 2021 Sep 18;8(9):116. doi: 10.3390/jcdd8090116.) and possible relationship with depressive disorders (see J Am Heart Assoc. 2019 Feb 19;8(4):e010825. doi: 10.1161/JAHA.118.010825. ). Please define better the population, describing if patients had MINOCA or classical myocardial infarction due to coronary atherosclerotic disease.

4) Please better discuss the possibile pathophysiological mechanisms that relates CRP with depressive disorders in patients with myocardial infarction

Author Response

Dear Editors and Reviewers,

We thank you and the Reviewers for the helpful advices to improve our manuscript. We now included recent and relevant paper(s) from a range of sources in the Reference list. The latter and changes we made to the manuscript in response to the Reviewers’ comments are highlighted in red.

Comments from the reviewer #1

Comment 1
There is a lack of a real control group. How can you affirm that patients with depression have a slower decline of CRP compared to patients without depression?

We thank the Reviewer 1 to point to the clarity of the manuscript and now include the Figure 2 showing a graph to illustrate the difference of CRP between participants with a history of depressions and those without depression.

Comment 2
Add an algorithm with study design

We thank the Reviewer 1 for the suggestion and now include the algorithm with inclusion and exclusion criteria (Figure 1)

Comment 3

Depressive disorders are a major risk factor for microvascular dysfunction and MINOCA. Please discuss the role of microcirculation in ischemic heart disease predisposition (see J cardiovasc Dev Dis 2021 Sep 18;8(9):116. doi: 10.3390/jcdd8090116.) and possible relationship with depressive disorders (see J Am Heart Assoc. 2019 Feb 19;8(4):e010825. doi: 10.1161/JAHA.118.010825. ). Please define better the population, describing if patients had MINOCA or classical myocardial infarction due to coronary atherosclerotic disease.

We thank the Reviewer 1 for the valuable suggestion and references that are now included in the discussion section.

Comment 4

Please better discuss the possible pathophysiological mechanisms that relates CRP with depressive disorders in patients with myocardial infarction

We thank the Reviewer 1 for the suggestion and added the following information to the discussion on possible pathophysiological mechanisms.

Our study was not designed to examine direct pathophysiological mechanisms that might link a history of depression with CRP levels in patients with acute MI. However, dysfunction in the autonomic nervous system (e.g. vagal withdrawal) [ref] and the hypothalamic-pituitary adrenal axis (e.g. low cortisol response) [13] are candidate mechanisms, both of which may result in failure to confine inflammatory activity. 

Reviewer 2 Report

Bielas et al. performed an analysis on association between a self-reported history of depression and a decrease of CRP concentration at 3-month follow-up in patients with acute myocardial infarction (MI) enrolled in the MI-SPRINT study. The authors concluded that history of depression may be associated with persistent elevation in CRP levels after acute MI.

This manuscript addresses an interesting topic, especially that low-grade inflammation manifested by elevated CRP concentration which persists after hospital discharge post-acute MI has been previously shown to be associated with adverse long-term outcome. Both the rationale and scientific content of the manuscript are valuable. However, I have several concerns that, in my opinion, deserve special attention in the revision. Specifically, I listed the most important of them below.

The history of depression was self-reported based only on a patient response “yes/no” to question about a presence of depression whenever in patient’s life. This information was acquired during an interview within first 48 h of hospitalization, just after admission for acute MI. This approach for diagnosing depression can be considered as a significant weakness of the study and requires more attention (e.g., in the study limitations) and clarification. I suggest adding a separate subsection devoted to this issue in Materials and Methods section which should provide more detailed description and validation of authors’ approach with well-defined inclusion criteria for this analysis and relevant references. Also, the title of the manuscript should include this information.

The study population included patients with different types of MI (i.e., STEMI and NSTEMI) which are known to be associated with different intensity of inflammatory activation in the course of acute MI. This issue needs further clarification because it can be considered as a study limitation.

The baseline characteristics of study population (Table 1) lacks an important information on comorbidities (such as diabetes mellitus, hypertension, prior heart failure, prior MI, etc.) which can be considered as potential significant covariates but was not included in the multivariate analysis. Additionally, there is no data about the angiographic indicators of reperfusion success, therapy, as well as prevalence of post-MI left ventricular dysfunction and remodelling, and heart failure, which have been previously shown to be associated with persistent elevation in CRP concentration after MI, especially STEMI, which was the main type of MI in the study presented in this manuscript. Omitting these aspects in the manuscript could affect the results of the analysis performed by the authors, so can be considered as a significant study limitation and weakness of the manuscript. The authors should provide missing data and include this data in multivariate analysis, especially that the authors stated that “None of the established risk factors of CVD, but a positive history of depression, was consistently identified as a significant predictor of persistently increased CRP levels in both regression models of the current study” (L. 221-223). However, the authors did not include in this analysis the factors associated specifically with the MI complications which may affect CRP levels. Additionally (especially if data are not available), this topic requires more attention in the Discussion section and study limitations. I suggest adding a separate paragraph devoted to this topic and including relevant references. Example relevant references that would fit in this context are:

Abbate, A.; Toldo, S.; Marchetti, C.; Kron, J.; Van Tassell, B.W.; Dinarello, C.A. Interleukin-1 and the Inflammasome as Therapeutic Targets in Cardiovascular Disease. Circ. Res. 2020, 126, 1260–1280.

Swiatkiewicz, I. et al. Enhanced inflammation is a marker for risk of post-infarct ventricular dysfunction and heart failure. Int. J. Mol. Sci. 2020, 21, 807.

Everett, B.M.; Cornel, J.H.; Lainscak, M.; Anker, S.D.; Abbate, A.; Thuren, T.; Libby, P.; Glynn, R.J.; Ridker, P.M. Anti-Inflammatory Therapy With Canakinumab for the Prevention of Hospitalization for Heart Failure. Circulation 2019, 139, 1289–1299.

Toldo, S.; Abbate, A. The NLRP3 inflammasome in acute myocardial infarction. Nat. Rev. Cardiol. 2018, 15, 203–214.

Fanola, C.L., et al. Interleukin-6 and the Risk of Adverse Outcomes in Patients After an Acute Coronary Syndrome: Observations From the SOLID-TIMI 52 (Stabilization of Plaque Using Darapladib-Thrombolysis in Myocardial Infarction 52) Trial. J. Am. Heart Assoc. 2017, 6, e005637.

ÅšwiÄ…tkiewicz, I., et al. I., Value of C-reactive protein in predicting left ventricular remodelling in patients with a first ST-segment elevation myocardial infarction, Med. Inflamm. 2012; 2012: 250867; doi:10.1155/2012/250867.

Suleiman, M., et al. Early inflammation and risk of long-term development of heart failure and mortality in survivors of acute myocardial infarction—predictive role of C-reactive protein. J Am Coll Cardiol. 2006;47:962–8.

The structure of the manuscript is not well organized and needs extensive improvements. For example, the aim of study should be explicitly defined in the last paragraph of the Introduction. Also, the Materials and Methods section is vague and needs improvements, especially in terms of precise description of the methods that were used in the study and providing the definitions of outcomes. For example, the inclusion criteria should be well-defined. The exclusion criteria do not include inflammatory diseases, which could affect CRP levels – this issue should be addressed in the manuscript. The tables with baseline characteristics and the results of multivariate analysis should be included in the main text of the manuscript. Also, while the authors indicated in the Discussion section that further research is needed in terms of associations between CRP and depression, an extension of this part of manuscript to specify future directions and broader clinical implications of future research in this field is required because this topic is not clear enough in the present manuscript. I suggest adding a short relevant paragraph. The Abstract section also requires further attention and some improvements following my other comments. The Conclusions section should be a separate section.

The scientific writing requires further attention because some statements can be made clearer to provide better understanding of specific message or content. The language and style also require checking and improvements.

Author Response

Dear Editors and Reviewers,

We thank you and the Reviewers for the helpful advices to improve our manuscript. We now included recent and relevant paper(s) from a range of sources in the Reference list. The latter and changes we made to the manuscript in response to the Reviewers’ comments are highlighted in red.

Comments from the reviewer #2

Comment 5

This approach for diagnosing depression can be considered as a significant weakness of the study and requires more attention (e.g., in the study limitations) and clarification.

In the limitations of the study, we extended on this issue stating that a structured clinical interview would have yielded greater confidence in the diagnosis of lifetime depression.

Comment 6

I suggest adding a separate subsection devoted to this issue in Materials and Methods section which should provide more detailed description and validation of authors’ approach with well-defined inclusion criteria for this analysis and relevant references. Also, the title of the manuscript should include this information.

We clarified in the Materials and Methods section that depression history was by patients’ self-report and added this information to the title. In addition, following comment 2 by Reviewer 1, we added as flow chart of patient recruitment into the study (Figure 1).

Comment 7

The study population included patients with different types of MI (i.e., STEMI and NSTEMI) which are known to be associated with different intensity of inflammatory activation in the course of acute MI. This issue needs further clarification because it can be considered as a study limitation.

We thank the Reviewer 2 to point to the clarity of the manuscript and now include the Figure 2 showing a graph to illustrate the impact of depression history on log CRP in the follow-up of participants with STEMI and NSTEMI. As can be seen in Figure 2, the relation between a history of depression and less of a decrease in log CRP levels was similar in patients with STEMI and those with non-STEMI.

Comment 8

The baseline characteristics of study population (Table 1) lacks an important information on comorbidities (such as diabetes mellitus, hypertension, prior heart failure, prior MI, etc.) which can be considered as potential significant covariates but was not included in the multivariate analysis.

We thank the Reviewer 2 for strengthening point 4 of our limitation section in the discussion. We edited/extended the limitations as follows:

 Fourth, covariates were selected based a priori assumptions on associations with CRP levels and were limited in number to avoid unstable coefficients due to model overfitting, so residual confounding by variables which were not controlled for remains a possibility. For instance, we were unable to additionally control in regression models for diabetes, prior heart failure, prior MI, and elevated triglycerides and blood pressure.

Comment 9

Additionally, there is no data about the angiographic indicators of reperfusion success, therapy, as well as prevalence of post-MI left ventricular dysfunction and remodelling, and heart failure, which have been previously shown to be associated with persistent elevation in CRP concentration after MI, especially STEMI, which was the main type of MI in the study presented in this manuscript. Omitting these aspects in the manuscript could affect the results of the analysis performed by the authors, so can be considered as a significant study limitation and weakness of the manuscript.

We thank the Reviewer 2 for the addition to point 4 of our limitation section and now state:

Similarly, only LVEF at the time of acute MI was available to be included as a covariate, although other cardiac-disease related measures have also been shown to be associated with persistent elevation in CRP concentration after MI, including angiographic indicators of reperfusion success, therapy, as well as prevalence of post-MI left ventricular dysfunction and remodelling, and heart failure.

Comment 10

The authors should provide missing data and include this data in multivariate analysis, especially that the authors stated that “None of the established risk factors of CVD, but a positive history of depression, was consistently identified as a significant predictor of persistently increased CRP levels in both regression models of the current study” (L. 221-223). However, the authors did not include in this analysis the factors associated specifically with the MI complications which may affect CRP levels.

We thank the Reviewer 2 for the clarification that our wording was misleading. We changed “established” to “examined”.

Comment 11

Additionally (especially if data are not available), this topic requires more attention in the Discussion section and study limitations. I suggest adding a separate paragraph devoted to this topic and including relevant references.

Please see our extensions on study limitations above (Comments 8 & 9). In the Discussion section/conclusions, we now additionally mention that “Larger studies allowing for extensive adjustment of correlates of CRP post-MI may further substantiate an independent relationship between lifetime depression and persistently elevated CRP levels post-MI.

Comment 12

We thank the Reviewer 2 for this suggestion and now included the following references in the discussion section:

Abbate, A.; Toldo, S.; Marchetti, C.; Kron, J.; Van Tassell, B.W.; Dinarello, C.A. Interleukin-1 and the Inflammasome as Therapeutic Targets in Cardiovascular Disease. Circ. Res. 2020, 126, 1260–1280.

Swiatkiewicz, I. et al. Enhanced inflammation is a marker for risk of post-infarct ventricular dysfunction and heart failure. Int. J. Mol. Sci. 2020, 21, 807.

Darapladib-Thrombolysis in Myocardial Infarction 52) Trial. J. Am. Heart Assoc. 2017, 6, e005637.

ÅšwiÄ…tkiewicz, I., et al. I., Value of C-reactive protein in predicting left ventricular remodelling in patients with a first ST-segment elevation myocardial infarction, Med. Inflamm. 2012; 2012: 250867; doi:10.1155/2012/250867.

Suleiman, M., et al. Early inflammation and risk of long-term development of heart failure and mortality in survivors of acute myocardial infarction—predictive role of C-reactive protein. J Am Coll Cardiol. 2006;47:962–8.

Comment 13

(…) the aim of study should be explicitly defined in the last paragraph of the Introduction. Also, the Materials and Methods section is vague and needs improvements, especially in terms of precise description of the methods that were used in the study and providing the definitions of outcomes. For example, the inclusion criteria should be well-defined. The exclusion criteria do not include inflammatory diseases, which could affect CRP levels – this issue should be addressed in the manuscript.

Thank you for bringing up these important points.

We clarified the aim of the study as follows:

…the aim of this study was to test the hypothesis of a direct relationship between a history of lifetime depression and persistent elevation in CRP across risk categories between hospital admission due to MI (when CRP levels reflect an acute phase response) and three months later (when CRP levels reflect low-grade inflammation).

We improved the Materials and Methods section in several places, also including a flow chart of patient recruitment (Figure 1).

We now mention in the limitations that inflammatory diseases were not an exclusion criterion.

Comment 14

The tables with baseline characteristics and the results of multivariate analysis should be included in the main text of the manuscript. 

We will address this valuable suggestion according to the journal’s requirements.

Comment 15

Also, while the authors indicated in the Discussion section that further research is needed in terms of associations between CRP and depression, an extension of this part of manuscript to specify future directions and broader clinical implications of future research in this field is required because this topic is not clear enough in the present manuscript. I suggest adding a short relevant paragraph.

We extended on this important issue in the discussion.

Comment 16

The Abstract section also requires further attention and some improvements following my other comments.

We edited the abstract accordingly.

Comment 17

The Conclusions section should be a separate section.

Done, as suggested by the Reviewer.

Comment 18

The scientific writing requires further attention because some statements can be made clearer to provide better understanding of specific message or content. The language and style also require checking and improvements.

We have edited the entire text extensively and hope that the manuscript now reads more clearly and with more focus.

Round 2

Reviewer 1 Report

All the comments have been correctly answered

Reviewer 2 Report

The authors adequately addressed my comments.